# The Aggressive Diabetic Kidney Disease in Youth-Onset Type 2 Diabetes: Pathogenetic Mechanisms and Potential Therapies

**DOI:** 10.3390/medicina57090868

**Published:** 2021-08-25

**Authors:** Michela Amatruda, Guido Gembillo, Alfio Edoardo Giuffrida, Domenico Santoro, Giovanni Conti

**Affiliations:** 1Unit of Pediatric Nephrology with Dialysis, AOU Policlinic G Martino, University of Messina, 98125 Messina, Italy; michela.amatruda@gmail.com; 2Unit of Nephrology and Dialysis, Department of Clinical and Experimental Medicine, University of Messina, 98125 Messina, Italy; guidogembillo@live.it (G.G.); alfiogiuffrida91@libero.it (A.E.G.); santisi@hotmail.com (D.S.); 3Department of Biomedical and Dental Sciences and Morpho-functional Imaging, University of Messina, 98125 Messina, Italy

**Keywords:** diabetic kidney disease, youth-onset type 2 diabetes mellitus, diabetic nephropathy, D2M pathophysiology and novel treatments

## Abstract

Youth-onset Type 2 Diabetes Mellitus (T2DM) represents a major burden worldwide. In the last decades, the prevalence of T2DM became higher than that of Type 1 Diabetes Mellitus (T1DM), helped by the increasing rate of childhood obesity. The highest prevalence rates of youth-onset T2DM are recorded in China (520 cases/100,000) and in the United States (212 cases/100,000), and the numbers are still increasing. T2DM young people present a strong hereditary component, often unmasked by social and environmental risk factors. These patients are affected by multiple coexisting risk factors, including obesity, hyperglycemia, dyslipidemia, insulin resistance, hypertension, and inflammation. Juvenile T2DM nephropathy occurs earlier in life compared to T1DM-related nephropathy in children or T2DM-related nephropathy in adult. Diabetic kidney disease (DKD) is T2DM major long term microvascular complication. This review summarizes the main mechanisms involved in the pathogenesis of the DKD in young population and the recent evolution of treatment, in order to reduce the risk of DKD progression.

## 1. Introduction

Diabetes mellitus (DM) is a chronic metabolic disease of both adults and children, and it is associated with long-term complications and high rates of mortality [1,2]. Its prevalence has been increasing, especially in young patients [3]. Type 1 diabetes mellitus (T1DM), also historically known as juvenile or insulin-dependent diabetes, is the most predominant type of DM in children and adolescents, being caused by insufficient pancreatic insulin production [4]. Multiple genetic and environmental factors found in variable combinations in T1DM individual patients [5].

In contrast, Type 2 Diabetes Mellitus (T2DM) or Type 2 Diabetes (T2D) is common in adults and is caused by insulin resistance [6]. T2DM is a complex metabolic disease with heterogeneous risk factors and complications; it consists of a combination of genetic predisposition, diet, or physical inactivity [7]. In the last decades, the prevalence of T2DM is becoming more common than T1DM, due to rising childhood obesity [8]. Recent data found that the highest prevalence rates of youth-onset T2DM were recorded in China (520 cases/100,000) and in the United States (212 cases/100,000), being the lowest in Denmark (0.6 cases/100,000) and Ireland (1.2 cases/100,000). However, specific epidemiological data for juvenile onset T2DM are currently scarce and considerably varied among different countries [9]. It is necessary to find uniform diagnostic criteria and finer screening strategies to reduce these differences [10].

The nature of T2DM is aggressive in young age, presents a sex-related variability, and is associated with unfavorable and early cardiometabolic risk factors related to obesity [11]. The Search for Diabetes in youth (SEARCH) and Treatment Options for Type 2 Diabetes in Adolescents and Youth (TODAY) studies compared young people with T2DM to healthy controls and demonstrated a higher prevalence of cardiovascular disease (CVD) and diabetic kidney disease (DKD) risk factors in the T2DM group [12,13]. Therefore, the main microvascular diabetic complication is DKD, which develops in about 25–40% of T2DM patients [14] and it is associated to a more rapid progression and poor prognosis in the long term [15,16]. In DKD subjects, early-onset T2DM is also an independent risk factor of end stage renal disease (ESRD) [17]. Faster progression to kidney damage is due to a poor glycemic control and the coexistence of multiple risk factors such as hyperglycemia, obesity, dyslipidemia, insulin resistance, hypertension, elevated serum uric acid, female sex, and inflammation [18].

The natural history of DKD begins with renal hyperfiltration, characterized by an increase in the glomerular filtrate rate (GFR) between 120 mL/min and 150 mL/min/1.73 m^2^, renal hypertrophy, and, sometimes, microalbuminuria [19]. Hyperfiltration is the first step of DKD, being primarily caused by obesity and impaired glucose tolerance [20]. The increase of intraglomerular pressure causes hyperfiltration, leading to mesangial expansion and glomerular basement membrane thickening [21]. These structural changes of the kidney are typical of nephropathy (stage 1) and are already present after 1.5–5 years from diabetes onset. Therefore, it represents a critical interval for risk factor reduction and prevention of the rapid and aggressive progression of DKD [22]. These changes are often reversible in the first years of the disease, depending on adequate metabolic compensation and normalization of GFR. Stage 2 DKD presents with glomerular lesions on biopsy and a GFR of 60–89% but no clinical signs; at this step, blood pressure and urine albumin excretion remain in the normal range. After 5–10 years from the onset of diabetes, the early DKD stage occurs (stage 3), characterized by microalbuminuria, defined as a urine albumin excretion of 30–299 mg/day [23]. Microalbuminuria could be considered a predictive element and/or biomarker of kidney damage; it may be already present in young people with a mean duration of T2D of only six months as a consequence of longstanding periods of misdiagnosis [24,25]. The regression to normoalbuminuria in patients with adequate metabolic control, normal blood pressure, and favorable lipid profile may occur [26]; in contrast, in case of poor glycemic control, hypertension, and dyslipidemia, further progression of damage occurs. Irreversible alterations related to the increased arterial blood pressure cause progression to the following stage (called “overt diabetic nephropathy or stage 4”). The main structural changes stage 4 related to kidney biopsy are diffuse and nodular glomerulosclerosis and arteriolar hyalinosis and, it is described by the presence of albumin ≥ 300 mg/day and followed by accelerated GFR decline. Eventually, the progressive loss of function leads ESRD, the stage 5 of DKD with GRF of under 15% [27].

Recently, the American Diabetes Association (ADA) has underlined the clinical importance of “persistent albuminuria” as a risk factor for nephropathy and CVD. Approximately 50% of patients with persistent albuminuria develop ESRD over 7–10 years after the onset of DM [28]. The rapid decline of GFR is equally important to microalbuminuria as a risk factor for DKD progression. GFR below 60 mL/min/1.73 m^2^ indicates the loss of half renal function with a small number of functioning glomeruli. For this reason, an annual measurement of estimated GFR is recommended to monitor and prevent ESRD [29].

Moreover, it is of pivotal importance distinguishing between DM-related and non-DM related forms of kidney disease in DM subjects, as different diseases require different clinical management, especially for the most aggressive form of kidney impairment [30].

However, it is mandatory to point out that a significant percentage of patients do not follow the classic trajectory of hyperfiltration, microalbuminuria, proteinuria, and decline in GFR [31]. Accordingly, an accelerated GFR decline has been described in T2DM patients, even in the absence of proteinuria [32]. Therefore, it is necessary to identify an alternative biomarker to microalbuminuria that can predict early risk of DKD and prevent permanent renal injury and refractoriness to common therapeutic options in T2DM children.

This article provides an overview of the main pathogenetic mechanisms involved in T2D aggressive nephropathy, as well as the currently available management options to reduce DKD risk. The discussion begins by describing correlation between DKD and CV risk and then discusses the pathophysiology and novel therapeutic targets to improve glycaemic control, hypertension, the lipid profile, and the progression of renal disease. These data are useful for research and clinical practice and provide novel and recent information on the new therapeutic target discovery and the development of potential novel therapies slowing progression of the aggressive DKD in youth onset type 2 diabetes.

## 2. Methodology

### 2.1. Literature Search

A literature review following PRISMA [33] (Preferred Reporting Items for Systematic Reviews and Meta-Analyses) guidelines was conducted in the PubMed gateway of the MEDLINE database and Clinicaltrials.gov. This research identified all articles published in English, in peer-reviewed journals from January 2000 to May 2021.

We used the following terms and mesh headings for the bibliography search: (“Diabetic Kidney Disease” OR “Diabetic Nephropathy”) AND (“T2DM” OR “Diabetes Mellitus” OR “Diabetic”) AND (“Young” OR “Children” OR “Adolescence” OR “Adolescents”). The reference list of each article and gray literature was scrutinized for additional relevant articles. We collected search results in an Endnote library. 

### 2.2. Study Selection Criteria and Procedure

DKD definition was indicated as evidence of kidney damage related to DN, defined as the presence of an alteration of kidney function in diabetic patients, diagnosed by an altered estimated glomerular filtration rate (eGFR) for at least three months, provided that other causes of CKD are excluded (e.g., rapid decreases of eGFR and/or significant increases in albuminuria, refractory hypertension, hematuria, nephrotic or nephritic syndromes, among others) [34,35]. CKD, in turn, was indicated according to the National Kidney Foundation-Kidney Disease Outcomes Quality Initiative (NKF KDOQI) guidelines by the presence of a reduced GFR < 90 mL/min/1.73 m^2^ or the persistence of hyperfiltration and/or urinary abnormalities (such as pathological albuminuria, proteinuria or hematuria) in subjects with GFR ≥ 90 mL/min/1.73 m^2^ [36].

The definition of Young-onset T2DM was defined as T2DM in people aged <40 years [37]. By screening abstracts, two reviewers (G.G. & A.E.G.) independently assessed the eligibility of each article. Disagreements were resolved by discussion between the two reviewers or by consulting a third author (G.C.). We only included studies meeting the following criteria: (1) Interventions were therapies for T2DM. (2) Outcomes of study must include pharmacological therapies potentially useful to slow the progression of the disease in patients with Youth-Onset T2DM. Exclusion criteria were observational studies, case reports, letters to editors, editorials, posters, studies not providing short- or long-term data on the outcomes of interest. Studies were considered regardless of drugs dosage or if the comparator was a placebo or standard treatment.

### 2.3. Search Results

Figure 1 shows the flow diagram of the study selection process. From 1095 unique identified studies identified (1158 including duplicates), we find 82 additional records through personal research and citation searching.

We identified 194 studies for the full text screening. After further screening full texts, 55 studies were identified for our review. Main characteristics of the included studies are described in Figure 1.

## 3. Correlation between DKD and CV Risk

DKD is associated with a very high CV (CV) risk and greater mortality rate than diabetic patients without renal damage [38]. The increase in mortality is a result of the higher incidence of major fatal and non-fatal CV events (MACEs) and worsening of the GFR [39]. The cardiorenal risk has been defined as the risk of progression of CV events (MACE or heart failure) and DKD that remains after the optimal glycemic control in T2DM [40]. 

Several studies demonstrated that nephropathy and CV risk is closely associated with reduced GFR and increased early markers for kidney damage as albuminuria excretion. This important cardiorenal risk marker determines inflammatory phenomena resulting in overall glomerular damage [41,42]. Even at the lower concentrations represented by microalbuminuria, albuminuria is considered an expression of endothelial damage and consequent increased systemic vascular permeability involving both glomerular and tubular structure and, at the same time, the endothelium that increased CV risk [43]. DKD perturbs the hemodynamic stability of CV system, increasing the risk of acute coronary syndromes [44].

A determinant link between DKD and CV risk is represented by the systemic vascular damage consequent to microalbuminuria [45,46]. The impact of albuminuria reduction on the CVCV outcome through the use of different drug classes is the most interesting topic of recent research [47]. 

A central role in the onset of proteinuria is the alteration of the glomerular endothelial glycocalyx; it represents an early event of diabetic damage due to the effects of hyperglycemia and the increased permeability of the glomerular filtration barrier of serum proteins [48]. Mitochondrial dysfunction of podocytes also plays a key role in the loss of the selective permeability of the glomerular membrane and the development of microalbuminuria, and it could be triggered by various mechanisms involving intracellular homeostasis [49,50]. In the pathogenesis of albuminuria in DKD, the reduced tubular reabsorption of albumin has been described and associated with the degree of renal dysfunction [51]. Nevertheless, the association of increased albuminuria with all-cause mortality and CVCV mortality in the general population remains controversial and poorly understood and at high risk of being undertreated. Recently, several potential biomarkers seem to play a role in the progression of DKD. Although albuminuria/proteinuria remain the most used markers of DN, the role of oxidative stress in the development of vascular damage in T2DM has been described, but it is not clear how its products could be included among renal markers in both clinical practice and clinical trials [52,53]. 

Several studies have shown how proteinuria and micro/macroalbuminuria are prognostic factors for the onset and worsening of DKD and CV disease. Minutolo R. et al. have recently been conducted a pooled analysis of four cohort studies, enrolling chronic kidney patients (CKD) treated with RAS inhibitors therapy to compared the risk of all-cause mortality, fatal and non-fatal CV events and ESRD between diabetic and non-diabetic patients stratified by proteinuria level. They concluded that non-proteinuric DKD patients do not have higher cardiorenal risk when compared with their counterpart non-diabetic patients. In contrast, moderate proteinuria entails a higher CV morbidity and mortality only in DKD patients, whereas severe proteinuria levels modulate the ESRD risk independent of diabetes. This study provides novel information on CV and renal prognosis in DKD patients referred to nephrology clinics; in fact, the effect of diabetes only emerges in the presence of moderate proteinuria and the finding was that proteinuria was a major modifier of prognosis [54]. 

In DKD patients, the therapeutic approach requires simultaneous treatment of global CVCV risk factors rather than a single risk factor to significantly improve above hard outcome with a short intervention, and a long durability of protection [55]. Nephropathy in Diabetes type 2 (NID-2) multicenter, cluster-randomized, open-label clinical trial demonstrates that in DKD patients at very high CVCV risk, intensive multifactorial treatment (MT) significantly reduces the risk of MACEs versus standard of care [56]. The factors correlated to the progression of nephropathy in diabetic patients are hypertension, poor glycemic control, dyslipidemia, proteinuria levels, obesity and smoking and glycemic control has long been though fundamental to diabetes management to reduce diabetes complications. Nevertheless, intensive glycemic control (IGC) has controversial effects in reducing the macrovascular complications associated with T2D. In conclusion, in DKD an effective therapy must not be limited to the management of hyperglycemia alone, but also of the other main risk factors to improve the mortality and CV outcome in diabetic patients [57]. 

## 4. Risk Factors for DKD in Youth with T2DM: An Epigenetic and Genetic Possible Link?

Young individuals with early onset of diabetes have a major risk for macrovascular and microvascular complications, secondary to longer disease exposure and potentially influenced by a genetic predisposition [58]. Juvenile T2D nephropathy occurs earlier in life compared to T1DM-related nephropathy of children or T2DM-related nephropathy of adults. The reason is that T2DM young people have multiple coexisting risk factors, including obesity, hyperglycemia, dyslipidemia, insulin resistance, hypertension, female sex, and inflammation. Youth onset T2DM patients also have a strong hereditary component often unmasked by social and environmental risk factors [59]. The most common comorbidities in children, especially in the USA, are obesity and increased body mass index (BMI). Their prevalence continues to escalate, and they are much more common in youth-onset T2DM than those with T1DM. Obesity is associated with insulin resistance (IR) and consequent poor glycemic control, defined as an HbA1c ≥ 9.5% [60]. In the initial pathogenesis of T2DM, pancreatic beta cells can increase the secretion of insulin and compensate IR maintaining good glycemic control. Then, pancreatic beta cells function declines, and the compensatory hyperinsulinemia is insufficient to compensate IR, resulting in hyperglycemia. The latter was reported in 27% of youth with T2DM, and it is more commonly observed among ethnic minorities and children with a family history of T2DM in relatives [61]. Markers of IR are acanthosis nigricans and polycystic ovarian syndrome (PCOS). Acanthosis nigricans is characterized by hyperpigmented patches in the intertriginous area, and it is a useful guide for screening children at risk for hyperinsulinemia. Some studies have shown that PCOS results from an excessive action of insulin on the ovary. It is a common disorder of premenopausal women characterized by hyperandrogenism, micro-polycystic aspect of the ovaries and anovulation. This syndrome has, therefore, reproductive, and metabolic morbidities [62]. 

Hypertension, defined as a condition with systolic or diastolic blood pressure over the 90th percentile, is present in youth with T2DM more often than those with T1DM. It appeared resistant to treatment and associated with male gender, increased BMI [63], left ventricular hypertrophy that appears after a few years from diagnosis [64]. Dyslipidemia is defined as an low-density lipoprotein (LDL) ≥ 130 mg/dL, high-density lipoprotein (HDL) < 40, hypertriglyceridemia, or use of LDL-lowering therapy. It has become a more frequent clinical condition in children and adolescents with T2DM than T1DM or adult-onset T2DM due to the increase in obesity prevalence and sedentary lifestyle. Dyslipidemia at this age range may be predominantly environmental (diet and lifestyle) and genetic related. It may be an extremely severe condition associated with major and early CVDs [65]. 

Hypertension, hyperinsulinemia, glucose intolerance, decreased HDL, and increased LDL and triglycerides are a cluster of metabolic abnormalities constituents of the “metabolic syndrome”, also called “syndrome X” or “syndrome of insulin resistance”. Children with T2DM have one or more features of metabolic syndrome. Furthermore, hypertension, dyslipidemia, and metabolic syndrome in T2DM children and adolescents are more aggressive and often diagnosed with delay, compared to T2DM adults or T1DM [66]. For this reason, T2DM children receive delayed treatment and become poorly responsive or resistant to medications. The resistance to treatments has been associated with a rapid loss of cell function and a worse disease progression [67]. 

Other risk factors for youth-onset T2DM are genetic predisposition, maternal history of diabetes, gestational DM and/or early puberty in girls. According to epidemiological studies, puberty is associated with insulin resistance because it has been demonstrated that GH secretion in a patient with a genetic predisposition of T2DM is probably responsible for the reduction of insulin action and consequent glucose intolerance at the mean age of this condition [68]. 

These mechanisms can be guided by genetic and non-genetic factors influencing gene expression. The non-genetic factors include the epigenetic causal interactions between genes and their products. Epigenetic factors have been also demonstrated to be associated with risk of both T1DM and T2DM development [69,70]. Changes related to puberty seems to have a possible direct influence on DKD lesions progression: the increasing of blood pressure, dysregulation of growth hormone-insulin-like growth factor I axis, overproduction of sex steroids [71].

Further epigenetic mechanisms contribute to DKD evolution, based on chromatin histone alterations, DNA methylation and non-coding RNAs, sustained by environmental factors [72]. Genome-wide association studies (GWAS) isolated a small number of genes, loci, and single nucleotide polymorphisms that can be related to DKD development [73,74]. Beyond the limit of the scarce number of gene-related factors that directly influence the DKD susceptibility and progression, their group demonstrated a promising role of genetic and epigenetic loci expression. Moreover, the Susztak laboratory developed kidney-specific epigenome maps and tissue-specific expression quantitative trait loci maps to isolate renal disease GWAS loci. A combination of the findings of these data sets’ maps showed consistent common genes involved in the development of diabetic complications [75,76]. 

A better understanding of these epigenetic and genetic mechanisms may lead to more effective strategies to treat DKD in young population at higher risk of disease progression.

## 5. Pathogenic Mechanisms

The pathophysiology of T2DM is characterized by an alteration in the equilibrium between insulin sensitivity and insulin secretion. T2DM results from the gradual decrement of β-cell activity and a crescent insulin resistance. Young patients with glucose imbalance demonstrate a higher impairment of insulin secretion compared with a reduction of insulin sensitivity [77]. 

A major cause of DKD development is represented by insulin resistance. This condition is clinically identified as the failure of exogenous/endogenous insulin to increment glucose uptake and utilization in as it does in an average population [78]. This resistance to the metabolic effect of insulin is tightly linked to other diseases such as dyslipidaemia, hypertension, and metabolic syndrome [79,80]. A dysregulation of insulin homeostasis contributes to lipoprotein and hepatic lipase inhibition, causing apolipoprotein and triglycerides to increase, particularly triglyceride-rich Very-low-density lipoprotein (VLDL) particles [81]. 

Patients presenting abnormal levels of these particles present a higher atherogenic plaque formation risk, with a decrease in HDL levels and a worsening of inflammatory status [82]. In fact, HDL presents a pivotal role in vascular health, preventing and reversing monocyte recruitment and activation into the arterial wall and regulating the expression of endothelial adhesion molecules [83]. 

These processes associated to insulin resistance contribute to a major risk of hypertension: a recent meta-analysis demonstrated that insulin resistance is independently associated with a higher risk of hypertension in the general population [84]. The link between insulin resistance and hypertension can be partially explained by the fact that this condition can induce renal sodium retention [85], contributes to overactivation of renin-angiotensin system [86,87], increases sympathetic nervous system activity [88], and stimulates peripheral and renal vascular resistance [89,90].

Atherosclerosis homeostasis can also be regulated by adipokines action, a family of adipose tissue-generated cytokines [91]. These molecules present a key role in inflammation mediation and the development of insulin resistance. Adipokines are defined as factors secreted by adipose tissue, exerting pro- and anti-inflammatory actions [92]. One example is represented by tumor necrosis factor alfa (TNF), which is overexpressed in the adipose tissue, promoting inflammation and antagonizing the insulin signaling [93,94]. Furthermore, interleukin-6 and interleukin-18 contribute to inflammatory upregulation, accelerating DKD evolution and kidney damage, directly affecting podocytes and worsening urinary albumin excretion rate [95,96]. 

Leptin represents another adipokine involved in metabolic syndrome and insulin resistance mechanisms. It controls appetite through an influence on the central nervous system [97] and seems to reverse insulin resistance and control dyslipidemia in experimental models [98]. Another major role in insulin resistance is represented by Neutrophil Gelatinase-Associated Lipocalin (NGAL), which is also linked to atherosclerosis and inflammatory processes [99]. At the same time, another adipokine, Plasminogen activator inhibitor 1 (PAI1), seems also correlated to T2DM development [100] and CV disease. 

Lack of adiponectin, another group of adipokines, is linked to vascular endothelial damage and insulin resistance, with a major risk of atherosclerotic plaque formation, playing a major role as anti-inflammatory hormone [101,102]. Lower adiponectin levels are negatively associated with the risk of obesity and metabolic syndrome [103,104]. In DKD patients, adiponectin showed a renoprotective function, translated in a reduction of albuminuria, glomerular hypertrophy, foot process effacement, regulation of mammalian target of rapamycin (mTOR) pathway, kidney inflammation mediators and modulates angiotensin II effects at renal tubular level [105,106]. 

The natural history of DKD shows an analogy with the progression of pancreatic islet β-cell failure in T2DM, such as hypertrophy of pancreatic islets, a proliferation of β-cells associated with inflammatory responses, and subsequent loss of β-cells by apoptosis and fibrosis of the pancreatic islets [107].

DKD is the consequence of several overlapping pathogenic mechanisms caused by early-onset type 2 diabetes. The main factors triggering the dysregulation of DKD pathways comprehend persistent hyperglycemia, obesity-related factors, β-cell failure, and insulin resistance. Both murine and human studies demonstrated that the increased cellular sugar concentration is the primary trigger leading to ESRD progression [108]. 

The Research Committee of the Renal Pathology Society proposed a pathologic classification of DKD depending on various degrees of disease severity [109] (Table 1).

In DKD patients, endothelial cells cannot downregulate their glucose transport, leading to an unbalanced intracellular glucose alteration [110]. 

High blood glucose leads to tissue damages by an increase of mitochondrial superoxide generation. Elevated intracellular superoxide activates poly (ADP-ribosyl)ation of glyceraldehyde-3-phosphate dehydrogenase (GAPDH) by poly(ADP-ribose) polymerase (PARP) that inhibits GAPDH [111,112]. The overproduction of glycolytic metabolites linked to Glyceraldehyde-3-P accumulation stimulates pro-oxidative mechanisms such as polyol and hexosamine [113] pathways: the accumulation of polyols in the mesangium provokes excessive matrix proteins deposition and mesangial expansion [114].

This dysregulated glycolytic metabolites production also promotes the formation of advanced glycation end-products species (AGEs) [115,116].

Specific AGEs receptors are present in both kidney podocytes and endothelial cells: intracellular formation of AGEs induce glomerular, endothelial dysfunction, and macrophage activation [117]. AGEs induce cell damage and alter the cellular proteins that regulate gene transcription, leading to cell cycle arrest and apoptosis. AGEs provoke a dysregulation between the production and removal of extracellular matrix proteins, causing a pathologic accumulation of collagens, fibronectins, and laminins [118]. Moreover, AGEs receptors promote inflammatory damage activating the JAK/STAT pathway. This mechanism stimulates the production of profibrotic cytokines and growth factors via the Receptor for Advanced glycation end-products species (RAGE) [119].

Hyperglycemia promotes the action of aldose reductase enzyme that reduces glucose in sorbitol, consuming nicotinamide adenine dinucleotide phosphate hydrogen (NADPH) [120]. NADPH is an essential cofactor for intracellular antioxidant regenerating, its decrement increases the intracellular oxidative stress [121].

The inflammatory response secondary to the hyperglycemic stress also leads to interstitial and glomerular macrophages infiltration. This accumulation is linked to a harmful cytokines cascade production, including TGF-β and necrosis factor tumor-alpha (TNF-a), leading to an accelerated declining of renal function [122].

At the renal level, hyperglycemia triggers Protein kinase C beta type (PKC-beta) and Protein kinase C delta type (PKC-delta). This process also stimulates the production of Interleukin-6 and Tumor Necrosis Factor-α from endothelium and mesangium [123,124]. 

PKC stimulation involves a considerable variety of pathogenic mechanisms: (1) Lower endothelial nitric oxide (NO) expression and increasing of endothelin-1 (ET-1), vasoconstrictor causing flow alterations; (2) Vascular endothelial growth factor (VEGF) over expression, with an increase in vascular permeability and angiogenesis; (3) TGF-β, fibronectin, and collagen increments, causing an extracellular matrix expansion and capillary occlusion; (4) PAI-1 growth that increases fibrinolysis and the risk of vascular damage; (5) Nuclear factor kappa-light-chain-enhancer of activated B cells (NF-kB) that promotes pro-inflammatory genes expression, promoting apoptosis and inflammatory process [125]; (6) NAD(P)H-NADPH oxidized increase resulting in reactive oxygen species (ROS) production [126,127,128]. 

Persistent hyperglycemia not only leads to PKC and AGEs-related inflammatory damage, but also several dangerous pathways, leading to a worsening of kidney impairment. A dysregulated glycemic control stimulates an overexpression of Janus kinase/signal transducers and activators of transcription (JAK/STAT) signaling pathway, transforming growth factor-beta1/SMAD pathways, Wnt/β-catenin signaling pathway, Integrins/integrin-linked kinase (ILK) signaling pathway, MAPKs signaling pathway, Jagged/Notch signaling pathway all driving to the formation of proinflammatory molecules, extracellular matrix deposition, and myofibroblast proliferation [129]. 

## 6. Prevention of DKD and Potential Therapies 

The prevention of DKD at a young age is an important challenge based on the protection of all those risk factors that contribute to the development and progression of this dreaded intravascular complication of diabetes. Often the onset of DKD occurs in about 7–10 years [130]. Therefore, the importance of early detection of these risk factors is essential to improve their quality of life. 

These risk factors consist of increased blood pressure, altered lipid profile, chronic state of hyperglycemia, and an unhealthy lifestyle [131], then we can act on them. The intervention on these factors reduces the risk of DKD progression, the deterioration of albuminuria, and renal impairment [132]. A rigorous control of the glycated hemoglobin (HbA1c) values in patients with DKD is essential to prevent the progression of it, also reducing CV events [133].

The ADVANCE study has highlighted this importance and how the control of serum glucose levels leads to a reduction in the onset of albuminuria. Regarding the control of hypertension, maintaining blood pressure under 145/80 mmHg leads to a lower incidence of DKD [134]. Experimental studies demonstrated how this control permits the reduction of proteinuria in diabetic mice, decreasing the glomerular damage [135]. 

Concerning lifestyle habits, exercise can improve other risk factors such as hypertension, dyslipidemia, insulin resistance, and CV risk in diabetic patients [136,137]. Further modifications on lifestyle are smoking cessation [138], a healthy diet constituted by lower intake of potassium, and salt [139,140]. The management of these risk factors related to their early recognition is a crucial challenge for nephrologists, reducing the incidence of DKD.

The therapeutic approach to DKD is characterized by a series of changes in life habits and, if necessary, a pharmacological treatment to improve glycemic control, hypertension, the lipid profile, and the progression of renal disease [141]. Among lifestyle changes are included the increase of physical activity, improvement in sleep restoration, and healthy nutrition [142]. A high protein intake (above 1.3 g/day) is correlated with an increased decline in kidney function, worsening of DKD with increased albuminuria and mortality. Even a restrictive sodium intake can enhance the anti-protective effects supported by ACEi [143]. A therapeutic scheme with metformin or orlistat is widely used in young obese patients and in the presence of an important state of insulin resistance [144]. These multifactorial interventions play a crucial role in glycemic control, reducing microalbuminuria and progression of overt DKD. 

In the last years, novel therapeutic targets, leading to the discovery of promising treatments, have been investigated [145]. 

*Sodium-glucose co-transporter 2 (SGLT2) inhibitors* are important emerging agents that blocks glucose and sodium reabsorption in the kidney. SGLT2 is the major sodium-glucose co-transporter, and it is located in the early proximal tubule. This important carrier leads to the reabsorption of glucose and sodium. Therefore, its block brings glycosuria and natriuria, which decreasing serum glucose and increases sodium delivery to the macula densa, with consequent reducing tubuloglomerular feedback activation and renal blood flow [146]. Among SGLT2i, Empagliflozin has been demonstrated to be well tolerated with a sensible reduction in HbA1c levels in T2DM and CKD [147]. Currently, the scientific community is waiting for results from Empa-Kidney trial, it should help us switch in the pediatric population in the coming years. 

*Receptor agonists of glucagon-like peptide-1 (GLP-1)* and *dipeptidyl-peptidase IV (DDP-IV) inhibitors* (or gliptins) belonging to the new hypoglycemic agents. These drugs are responsible for the incretin effect. GLP-1 leads to a fall in liver glucose production reducing appetite and body weight [148]. Further action is antihypertensive, increasing levels of atrial natriuretic peptide and improving endothelial cell function [149]. The inhibition of sodium–hydrogen exchanger 3 (NHE3), which is situated in the cells of the proximal convoluted tubule, contributes to an increase in natriuresis [150]. In animal studies, the use of these molecules leads to a reduction in the activation of the renin–angiotensin–aldosterone system (RAAS), which is one of the main stimuli for the onset of DKD. This mechanism appears to improve also glomerular hemodynamics [151]. The DPP-IV inhibits the enzyme dypeptilpeptidase responsible for the degradation of GLP-1. Besides the hypoglycemic function, there is also an anti-hypertensive action, and the contribution in maintaining an adequate lipid profile and reducing oxidative stress with an improvement in endothelial cell response [152]. It also appears to be safe in patients who perform dialysis treatment [153]. However, further research to confirm the clinical outcome in patients with DPP-IV inhibitors treatment are needed [154].

*Thiazolidinediones (TZDs) or Glitazones* are potent activators of nuclear receptor gamma (PPAR-γ), warranting an important hypoglycemic effect through an improved sensitivity to insulin [155]. The PPAR-γ are widely located in the adipose tissue, so a higher activity of these receptors with TZDs administration, is involved in the reduction of insulin-resistance [156,157]. This class of drugs shows pleiotropic effects beyond glucose-lowering. The pioglitazone seems to reduce the CV risk through an anti-atherogenic action on the vessel wall [158]. The main side effect of TZDs use is the gain of body weight either in monotherapy or in association with insulin, metformin, and sulfonylureas. This effect was limited to the association of TZDs with SGLT2i [159] or GLP1 [160]. More side effects include water retention [161] with increased risk of heart failure and osteoporosis [162]. For these reasons, the use of these drugs has been largely limited specially in juvenile diabetes, although other trials would be needed to clarify their actions in the pediatric cohort. 

Phosphodiesterase inhibitor (*Pentoxifylline*, PTF) reduces the production of tumor necrosis factor (TNF-α) with consequent inhibition of cell proliferation. These drugs inhibit the phosphodiesterase pathway, bringing to increase of intracellular cAMP. However, further studies to develop recommendations for its use are needed Vitamin D presents an important role in the anti-inflammatory mechanism, immune response modulation, and RAAS inhibition [163,164]. In DKD renal tubular epithelial cells, vitamin D receptor expression is downregulated. Pyridoxamine is a molecule belonging to the Vitamin B6 family and its activity is based on AGEs production and/or reducing oxidative stress. Further studies are needed, however, to prove efficacy in slowing kidney disease progression are necessary. This molecule seems to decrease the risk of intervertebral disc degeneration [165], an important risk factor that can involve the people affected by T2DM [166]. Pirfenidone, Bardoloxone, and APX-115 molecules active on AGEs production. *Bardoloxone* is a semi-synthetic product deriving from natural oleanolic acid. It is a potent activator of Kelch-like ECH-associated protein 1 (Keap1)/nuclear factor erythroid 2-related factor 2 (Nrf2) system leading to decrease the transcription of NF-kB and consequently the inflammation process [167,168]. The Nrf2 is located in several districts with the highest expression at the kidney level [169]. The activity of Nrf2 is reduced in CKD [170] and it is associated with GFR improvement [171]. *APX-115* is a drug belonging to NADPH oxidase (NOX) inhibitor [172] studied DKD mice.

Molecules with effects on inflammatory and fibrosis pathways are *Baricitinib* and *Ruboxistaurin.* Baricitinib, a selective JAK-1 and JAK-2 inhibitor. 

ACE inhibitor, angiotensin II receptor blocker (ARB), Nonsteroidal mineralocorticoid receptor antagonist (MRA), and Aliskiren are molecules that inhibit the RAS pathways. Hyperactivation of this via leads to an increase of intraglomerular pressure creating vasoconstriction at the level of efferent arterioles generating proteinuria, mesangial proliferation and activation of the inflammatory process with final result of fibrosis [173,174]. Among the most consolidated therapies for DKD is the use of ace inhibitors that are a blocker of the enzyme of conversion of angiotensin I in angiotensin II, representing a cornerstone blocking the renin angiotensin aldosterone system and limiting the previously seen damage cascade. A double block-therapy with telmisartan and ramipril showed a worsening of nephropathy and mortality [175]. Mineralocorticoid Receptor Antagonists (MRA) are widely used for DKD treatment. Finerenone and Esaxerenone are a new generation of nonsteroidal MRA that selectively block the receptor with a better safety profile without causing the side effects (hyperkalemia and renal dysfunction) typical of this pharmaceutic class [176,177]. Aliskiren is a direct renin inhibitor that it protects against diabetic kidney disease. 

Atresantan is a selective endothelial A receptor antagonist. Endothelin A is a powerful vasoconstrictor factor that increases due to several damage mechanisms such as higher level of glycemia, insulin resistance, and uncontrolled activity of RAAS. By contrast, non-selective endothelin receptor antagonist, also inhibiting the endothelial B receptor, increasing fluid retention with a greater incidence of heart failure [178]. 

Sulodexide (SXD) is glycosaminoglycan (GAG) with antithrombotic and profibrinolytic activity with renoprotective effect on diabetic patients. 

Monoclonal antibodies (mAbs) are new molecules active on the main protein responsible for DKD inflammation and fibrosis. Further studies to show their efficacy and safety in DKD treatment are needed to understand their use in real life. 

Stem cell therapeutic strategies are currently under investigation, but further and larger studies are needed to establish the safety, feasibility, tolerability, and long-term effect. 

Hypoxia-inducible factor prolyl hydroxylase inhibitor seems to open an important key in renal protection in DKD patients. 

Proprotein convertase subtilsin-kexin type 9 inhibitors (PCSK9i) are new lipid-lowering drugs (monoclonal antibodies) that block enzyme PCSK9. This is a liver protease that attacks LDL receptors, bringing their lysosomal destruction, leading to an increase in circulating LDL [179]. Several trials have shown a familial hypercholesterolemia and earlier cardiovascular disease in patients with activate mutation of PCSK9 [180,181]. The liver is not the only manufacturing site of this protease; other particular locations are gut and kidneys [182,183]. In patients undergoing peritoneal dialysis and with nephrotic syndrome, higher blood PCSK values have been highlighted which leads to an increase in LDL in this population [184].

Evolocumab and Alirocumab are the only two drugs that are FDA-approved and administered subcutaneously [185,186]. The PROFICIO trial has evaluated the safety of Alirocumab, showing the main side effect related to the use of this drug, nasopharyngitis [187]. In addition, the benefit to the lipid profile with the reduction of LDL values, guarantees a dyslipidemia control which is present in many patients with metabolic syndrome and DKD. Further studies could open up new scenarios on the use of these drugs in the management of metabolic syndrome, microalbuminuria and DKD.

Fibrates are drugs involved in the lipid and glucidic balance. Their action is based on activation of Peroxisome proliferator-activated receptor-α (PPAR- α), decreasing the triglyceride levels and consequently the atherosclerotic process [188,189]. These receptors are ubiquitously located. In the renal tissue, the PPAR-α is expressed in several districts like mesangium, proximal tubular cells and collecting duct [190,191,192]. 

These therapeutic approaches are summarized in Table 2.

## 7. Conclusions

In juvenile DKD, the contemporary presence of several comorbidities and the consequent risk of development aggressive nephropathy can determine a reduction in the quality of life and an increase in mortality rates, despite shorter disease duration than T1DM and comparable glycemic control [226]. 

Regarding those T2DM young patients at major risk of aggressive nephropathy, the study of a proper strategy to combine different hypoglycaemic drugs, alone or associated with insulin therapy, is of pivotal importance. 

The study of new therapeutic target discovery and the development of potential novel therapies is still under evaluation but the results from most recent studies are promising. An ideal therapy should guarantee a slowing progression of the diseases and maintain great flexibility in the timing of meals and daily activities for young patients, avoiding potential lethal hypoglycemic levels, thereby enhancing patients’ quality of life. 

## Figures and Tables

**Figure 1 medicina-57-00868-f001:**
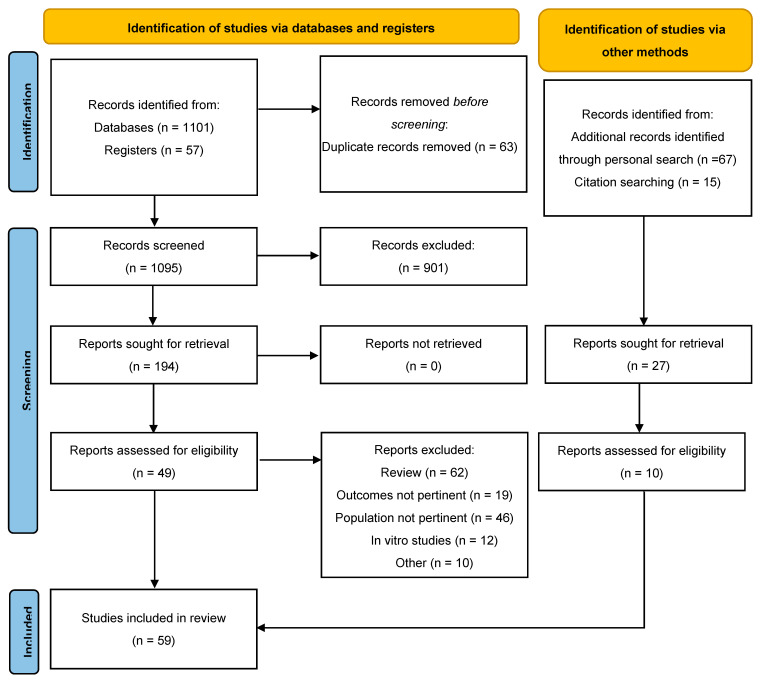
Flow diagram of the study selection process. From: Page MJ, McKenzie JE, Bossuyt PM, Boutron I, Hoffmann TC, Mulrow CD, et al. The PRISMA 2020 statement: an updated guideline for reporting systematic reviews. BMJ 2021;372:n71, doi:10.1136/bmj.n71. For more information, visit: http://www.prisma-statement.org/.

**Table 1 medicina-57-00868-t001:** Histological staging of diabetic glomerulopathy.

Class I	Class II	Class III	Class IV
GBM thickening alone, GBM > 430 nm in men and >395 nm in women.	Mesangial expansion present in >25% of the mesangium	Nodular sclerosis, characterized by the presence of Kimmelstiel–Wilson lesions but <50% diffuse global glomerulosclerosis	Advanced diabetic glomerulosclerosis, defined as >50% diffuse global glomerulosclerosis with or without nodules

Abbreviations: GBM = glomerular basement membrane.

**Table 2 medicina-57-00868-t002:** Therapeutic agents for DKD treatment: mechanisms of action and effects on renal outcomes.

Class of Drugs	Mechanism of Action	Effects on DKD	Ref.
SGLT2 INHIBITORS	Block the sodium-glucose co-transporter located in the proximal tubule avoiding a reabsorption of glucose and sodium	Reduction of microalbuminuria; Slower progression of kidney disease Decreased of HbA1C and serum glucose	[193,194]
GLP-1 AGONISTS	Increase the insulin production and decrease glucagon secretion	Improvement in glomerular hemodynamic reducing RAAS activation and DKD incidence	[195,196]
DPP IV INHIBITORS	Inhibit the dypeptilpeptidase responsible for degradation of GLP1, enhancing incretin effect	Improve endothelial function Reduce the progression of albuminuria	[197,198]
GLITAZONES	Activate of PPAR-γ improving insulin sensitivity	Decrease microalbuminuria and inflammation process	[199,200]
PENTOXIFYLLINES	Inhibit the phosphodiesterase increasing cAMP concentration	Reduce TNF-α Slow urinary albumin excretion Increase insulin sensitivity	[201,202]
VITAMIN D	Anti-inflammatory mechanism also modulating RAAS inhibition	Reduce proteinuria and renal disease progression	[203,204]
PYRIDOXAMINE	Belonging to the vitamin B6	Decrease AGEs production and oxidative stress	[205]
PIRFENIDONE	Blocks TGF-β production and TGF β-NADPH-induced ROS formation	Reduces the expression of TNF-α Decreases mesangial expansion and glomerulosclerosis showing antifibrotic effects Improves the GFR of 25% in patient with glomerulosclerosis	[206,207]
BARDOLOXONE	Activates of Keap1 and Nrf2 leading to decrease of NF-kb	Reduces inflammation process Increases glomerular filtration rate	[208,209]
APX-115	Inhibits NADPH oxidase	May have an important anti-inflammatory action (only experimental studies)	[210,211]
BARICITINIB RUBOXISTAURIN	Inactivate JAK-1 and JAK-2	Reduce albuminuria, DKD progression and inflammatory process Ruboxistaurin works as PKC-β inhibitor reducing albuminuria with normal GFR	[212,213]
ACE INHIBITORS ARB MRA RENIN INHIBITORS	Inhibit the RAAS pathways	Reduce proteinuria, mesangial expansion and DKD progression. Telmisartan and Ibersartan limit the transition from microalbuminuria to overt proteinuria about 60%. Moreover, 70% Spironolactone and Eplerenone reduce inflammation, albuminuria and fibrosis Finerenone shows a reduction of proteinuria, UACR and tubular damage showing also the CV protection Aliskiren decreases albuminuria and glomerulosclerosis limiting the TGF-β action and lipids accumulation	[214,215]
ATRESANTAN	Selectively antagonizes endothelial A receptor	Reduces insulin resistance Downregulates fibrosis, inflammation pathways, and DKD progression	[216,217]
SULODEXIDE	Glycosaminoglycan with antithrombotic and profibrinolytic activity	Reduces albuminuria and protects tubular cells from oxidative stress	[218,219]
MONOCLONAL ANTIBODIES (VPI-2690B, FG3019, LY3016859)	The first target is Vitronectin receptor The second target is tissue growth factor The third target is TGF-α	Decrease DKD inflammation and fibrosis (Only phase II studies)	ClinicalTrials.gov Identifier: NCT02251067 ClinicalTrials.gov Identifier: NCT01890265 ClinicalTrials.gov Identifier: NCT01774981
STEM CELLS	Mesenchymal stem cells	Reduce albuminuria, fibrosis and ICAM-1 (only experimental studies)	[220]
HYPOXIA-INDUCIBLE FACTOR INHIBITORS	Inhibit Hypoxia-inducible factor prolyl hydroxylase decreasing NF-kb	Downregulate renal fibrosis and inflammation process Reduce the tubulointerstitial sclerosis due to hypoxia exposure	[221,222]
PCSK9 INHIBITORS	Monoclonal antibodies that block PCSK9 enzyme decreasing LDL blood concentration	Reduce the microalbuminuria and metabolic syndrome Decrease	[223]
FIBRATES	Activate the PPAR-α decreasing triglyceride levels	Decrease albuminuria value and metabolic syndrome Show less CV risk and heart attack in patient with diabetes Limit the albuminuria onset and progression	[224,225]

Abbreviations: DKD = diabetic kidney disease; SGLT2 = sodium-glucose co-transporter 2; HbA1C = glycated hemoglobin; GLP1 = glucagon-like peptide-1; RAAS = renin-angiotensin-aldosterone system; DPP IV = dypeptilpeptidase IV; PPAR-γ = peroxisome proliferator-activated receptor gamma; TNF-α = tumor necrosis factor alfa; cAMP = cyclic adenosine monophosphate; AGEs = advanced glycation end-products; TGF-β = transforming growth factor beta; NADPH = nicotinamide adenine dinucleotide phosphate; ROS = reactive oxygen species; UACR = urinary albumin creatine ratio; GFR = glomerular filtration rate; CV = cardiovascular; Nrf2 = nuclear factor erythroid 2-related factor 2; NF-kb = nuclear factor kappa-light-chain-enhancer of activated B cells; PKC-β = protein kinase C-beta; JAK-1 = janus kinase 1; JAK-2 = janus kinase 2; ICAM-1 = intercellular adhesion molecule 1; PCSK9 = proprotein convertase subtilisin/kexin type 9; LDL = low-density lipoprotein; PPAR-α = peroxisome proliferator-activated receptor alfa.

## Data Availability

Data sharing is not applicable to this article as no new data were created or analyzed in this study.

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
