# Peer review of "The Aggressive Diabetic Kidney Disease in Youth-Onset Type 2 Diabetes: Pathogenetic Mechanisms and Potential Therapies"

_medicina, 2021, doi:10.3390/medicina57090868_

Round 1

Reviewer 1 Report

The paper is interesting. The topic is very hot. However, this reviewer raises some issues that must be addressed by authors.

1- This review in no way addresses the correlation between DKD and CV risk, which is a main issue in type 2 diabetes, in young patients too. Therefore, I recommend adding a paragraph on this issue.

a- Firstly, the authors should underline the strict relationship between AER and cardio-renal outcome in diabetic nephropathy observed in several studies (1- Nephrol Dial Transplant. 2018 Nov 1;33(11):1942-1949  doi: 10.1093/ndt/gfy032. 2- J Hypertens. 2006 Aug;24(8):1655-61 doi: 10.1097/01.hjh.0000239303.93872.31 3- Nephrol Dial Transplant. 2012 Jun;27(6):2269-74. doi: 10.1093/ndt/gfr644. 4- Diabetes Care 2006 Mar; 29 (3): 498-503. doi: 10.2337/diacare. 29.03.06.dc05-1776). This issue should be addressed by authors.

b- Very recently, mortality and MACEs has been investigated in a DKD population in the NID study (Cardiovasc Diabetol (2021) 20:145. doi: 10.1186/s12933-021-01343-1). This multicentric randomized trial originally demonstrated the ability of a multifactorial therapeutic approach to significantly improve above hard outcome with a short intervention (less than 3 years), moreover with a long durability of protection. Therefore, in DKD, an effective therapy must not be limited to the management of hypeglycemia alone, but also of the other main risk factors, as part of an integrated and multifactorial therapy, in order to improve the mortality and CV outcome in diabetic patients. This clinically very important issue should be commented in the discussion

2- A single table in a review seems too little.

3- References 2 and 27 refer to the same manuscript (Lascar N. et al).

4- A linguistic revision of the text by a native English speaker is suggested.

Author Response

Thank you for your precious comments. We believe that with your contribution our manuscript did gain more precision and clearness. We used the track changes modality of word to facilitate the recognition of our corrections.

We added a paragraph on the correlation between DKD and CV risk as you suggested, underlining the relationship between AER and cardio-renal outcome in diabetic nephropathy.  We also mentioned the relevant contribution of the NID study on mortality and MACEs in DKD patients. We also underlined the importance of insulin resistance, dyslipidemia, hypertension, and related pathophysiology mechanisms leading to a worsening of DKD.

We added also a table summarizing the therapeutic strategies for DKD treatment in these populations.

A linguistic revision by a native English collaborator and few more minor revisions have been performed.

Reviewer 2 Report

As shape: Please merge the paragraphs of 1-2 lines together, as the paragraphs to be more consistent. A paragraph is an idea, not a sentence! Revise the entire manuscript in this regard.

Abstract. It must be a compact, continuous paragraph, with no spaces between lines. Please remove them.

Introduction section.

  • It is much too short, not providing all aspects necessary for a good background in the topic.
  • It is very poor referenced.
  • The aim of the study does not highlight any new or special aspect which can recommend this paper to be published. Please improve it as so it must draw attention on the work/novelty of this paper.
  • L55-60 is Methodology part - I recommend to insert it in a new section, namely 2. Methodology. A PRISMA flow chart is recommended to be done please see both page at al. papers, where this type of graphic is very well described: Page, M.J.; McKenzie, J.E.; Bossuyt, P.M.; Boutron, I.; Hoffmann, T.C.; Mulrow, C.D.; Shamseer, L.; Tetzlaff, J.M.; Akl, E.A.; Brennan, S.E.; et al. The PRISMA 2020 statement: An updated guideline for reporting systematic reviews. Journal of Clinical Epidemiology 2021, 134, 178-189, doi:10.1016/j.jclinepi.2021.03.001. Page, M.J.; McKenzie, J.E.; Bossuyt, P.M.; Boutron, I.; Hoffmann, T.C.; Mulrow, C.D.; Shamseer, L.; Tetzlaff, J.M.; Moher, D. Updating guidance for reporting systematic reviews: development of the PRISMA 2020 statement. Journal of Clinical Epidemiology 2021, 134, 103-112, doi:10.1016/j.jclinepi.2021.02.003. Please renumber the other sections accordingly.
  • In this methodology part, please mention all criteria used for the papers' selection. You will see all of them in the PRISMA flow chart provided as model in the papers above recommended.

4. Pathogenic mechanisms: 

  • Please discuss the role of insulin resistance as a cause of DKD itself and as a cause of other diseases such as hypertension, dyslipidaemia that can increase the risk of DKD. I suggest as references Vesa, C.M.; et al. Current Data Regarding the Relationship between Type 2 Diabetes Mellitus and Cardiovascular Risk Factors. Diagnostics 2020, 10, 314. https://doi.org/10.3390/diagnostics10050314  to get a better perspective of this relationships between risk factors.
  • Please also discuss the role of adipokines in DKD pathogeny (Zaha et al. Influence of inflammation and adipocyte biochemical markers on the components of metabolic syndrome. Exp. Ther. Med., 20, 121-128. https://doi.org/10.3892/etm.2020.8663.
  • The hemodynamic effect of DKD on cardiovascular system and the risk of acute coronary syndromes must be detailed – I suggest Moisi et al. Acute Coronary Syndromes in Chronic Kidney Disease: Clinical and Therapeutic Characteristics. Medicina. 2020; 56(3):118. https://doi.org/10.3390/medicina56030118.
  • Original figures can much improve the visual impact of your paper and I consider this chapter to be very appropriate for insertion of figure/figures illustrating some pathogenetic mechanisms.  

5. Potential therapies:

  • I suggest to tabulate all data between L264-443. An exhaustive Table is much more relevant and easier to follow, especially when it comprises data from different references. The last column of the table must be named Ref. (from References), and all of them must be mentioned there, as well in brackets. Please check the style of the table recommended by the Instructions for authors. https://www.mdpi.com/journal/medicina/instructions.
  • Furthermore in this section, please discuss the beneficial roles of PCSK9 inhibitors in DKD, the role of fibrates in DKD, the role of mineralocorticoid receptor inhibitors with non-steroid structure (such as finernone) in the treatment of these patients.

A new section related to the prevention of DKD in youth by screening methods or by early identification of risk factors and their appropriate treatment must be introduced.

References. They must be cited in the text in brackets - i.e. [1], [1-4], not in superscript. Furthermore, in the current phase they are written at the final of the manuscript very carelessly, in a different style, not respecting any rules. The appearance of a work (in its entirety), as well as its content, is defining in its acceptance for publication. References must be set according to the journal's Instruction for authors. Please check https://www.mdpi.com/journal/medicina/instructions EndNoting them and choosing the MDPI style  is very useful.

Author Response

We would like to thank the reviewer for the comprehensive assessments, constructive criticisms, and valuable comments on our manuscript. Under your recommendations, we have revised our manuscript in response to your comments using the track changes modality.

According to the suggestion of the reviewer, and in order to clarify some relevant aspects of our paper, we merged paragraphs 1 and 2 together, expanding the concepts in the introduction section and adding more references. We also made the abstract section more compact.  We did not use the PRISMA flow chart for the nature of the reviewer itself, representing a narrative review and not a systematic review/meta-analysis.

We added a paragraph on CV role in DKD and improved the pathophysiology mechanisms section. We underlined the central role of insulin resistance as a cause of DKD and as a contribution to the risk factors main involved in its development and also commented on the impact of adipokines' action in DKD development mechanisms, both in pro and anti-inflammatory ways.

A table synthesizing the potential therapies has been added to give more importance to this essential topic. We also discussed, in the potential therapies section, the beneficial roles of PCSK9 inhibitors in DKD, the role of fibrates in DKD, the role of mineralocorticoid receptor inhibitors with non-steroid structure for this disease.

A new paragraph on DKD prevention in youth has been also added to our paper.

References have been cited in brackets and we conformed the citations following the journal’s instruction for authors.

Round 2

Reviewer 1 Report

The authors have addressed enough the issues raised by this reviewer.
I just raise a few comments.

1- There are still references that repeat themselves: again 3 and 57 (Lascar N. et al.) And 50 and 56 (Minutolo et al.). A careful check by the authors of all references is desirable.

2- I appreciate that the authors wanted to highlight the combinations, but they used 4 different font colors and the text does not appear well formatted, as does table 1. It would be helpful for authors to submit a clean and correctly formatted version of the manuscript.

3- Actually, in a review reviewers often suggest adding other references. I myself have suggested 5 new references, all in my opinion essential for this manuscript. But I was intrigued by the fact that in the first version of the manuscript there were 164 references, and in this second the references increased to 261. Have all the references added been suggested by the reviewers?

Author Response

To:

Medicina

Cover letter

Dear Editor in Chief,

We thank the editor and the reviewers for their comments and suggestions which enabled us to largely improve our manuscript.

Here enclosed you’ll find the “point to point” answers to comments and questions.

Reviewer 1

  • There are still references that repeat themselves: again 3 and 57 (Lascar N. et al.) And 50 and 56 (Minutolo et al.). A careful check by the authors of all references is desirable.

We thank the reviewer for reporting these claims to us. We have checked and modified all references according to the journal's instruction for authors.

  • I appreciate that the authors wanted to highlight the combinations, but they used 4 different font colors and the text does not appear well formatted, as does table 1. It would be helpful for authors to submit a clean and correctly formatted version of the manuscript.

We apologize. As suggested by the reviewer we send a version with corrections in Word review mode and a clean and correctly formatted version of the manuscript.

  • Actually, in a review reviewer often suggest adding other references. I myself have suggested 5 new references, all in my opinion essential for this manuscript. But I was intrigued by the fact that in the first version of the manuscript there were 164 references, and in this second the references increased to 261. Have all the references added been suggested by the reviewers?

As suggested by the reviewers we have increased the number of references in some cases for the reviewers' request, including the request of adding more specific paragraphs (eg "Correlation between DKD and CV risk" and "Materials and Methods" following PRISMA guidelines) and in response to other questions of reviewers. However, in the correction of the manuscript we noticed some errors in the numbering of the bibliography. Therefore, we send the corrected version with 252 total references.

Reviewer 2 Report

The authors made some improvements, but not the all requested. Once again: THE ASPECT of a paper and RESPECTING ALL the Instructions for authors is very important in its submission step. Please see below my concerns (which were also mentioned in my 1st report). I detailed them as follows:

  1. I understand that authors they did not want to respect one of my request (from my 1st report) regarding:
  • I recommend to insert it in a new section, namely 2. Methodology. A PRISMA flow chart is recommended to be done please see both page at al. papers, where this type of graphic is very well described: Page, M.J.; McKenzie, J.E.; Bossuyt, P.M.; Boutron, I.; Hoffmann, T.C.; Mulrow, C.D.; Shamseer, L.; Tetzlaff, J.M.; Akl, E.A.; Brennan, S.E.; et al. The PRISMA 2020 statement: An updated guideline for reporting systematic reviews. Journal of Clinical Epidemiology 2021, 134, 178-189, doi:10.1016/j.jclinepi.2021.03.001. Page, M.J.; McKenzie, J.E.; Bossuyt, P.M.; Boutron, I.; Hoffmann, T.C.; Mulrow, C.D.; Shamseer, L.; Tetzlaff, J.M.; Moher, D. Updating guidance for reporting systematic reviews: development of the PRISMA 2020 statement. Journal of Clinical Epidemiology 2021, 134, 103-112, doi:10.1016/j.jclinepi.2021.02.003. Please renumber the other sections accordingly.
  • In this methodology part, please mention all criteria used for the papers' selection. You will see all of them in the PRISMA flow chart provided as model in the papers above recommended.

but I still believe that it is the easiest way to present the selection of the references included in the study. I gave the references (which are not mines) exactly to help them in understanding a PRISMA flow chart (which is very easy to be done and followed. I apply it in all my review type papers.

2. Table 1 does not respect the Instructions for authors regarding settings. Please check https://www.mdpi.com/journal/medicina/instructions.

3. There remained in the text many very short paragraphs  that must be merged.

4. Please remove empty spaces between lines in the same section/subsection. Revise the entire manuscript in this regard. (i.e. section 5, 6, etc.)

5. In my previous report I also suggested :

  • I suggest to tabulate all data between L264-443. An exhaustive Table is much more relevant and easier to follow, especially when it comprises data from different references. The last column of the table must be named Ref. (from References), and all of them must be mentioned there, as well in brackets. Please check the style of the table recommended by the Instructions for authors. https://www.mdpi.com/journal/medicina/instructions.
  • Furthermore in this section, please discuss the beneficial roles of PCSK9 inhibitors in DKD, the role of fibrates in DKD, the role of mineralocorticoid receptor inhibitors with non-steroid structure (such as finernone) in the treatment of these patients.

The authors tabulated some of them but left also the text. Data must be provided in a single form (not in duplicate and I requested the table exactly for the reason to remove that the long list of potentially usable therapeutics (again with many empty lines and very short paragraphs for a single therapeutic). So, please complete the table with more details, and remove the text list of therapeutics/drugs. Table 2 does not respect the Instructions for authors as well. Furthermore, it can be "widened", adding more columns (if the authors consider it necessary), so that it occupies the entire width of the page.

The title of a Table must be inserted ABOVE the table.

Last column can be renamed in short as Ref. and is enough to mention there just the number of the reference used to support the text  - i.e. for SGL T2 [152-156]. The authors complicated themselves in this last column.

From my 1st report also. References. remained written at the final of the manuscript very carelessly, in a different style, not respecting any rules. The appearance of a work (in its entirety), as well as its content, is defining in its acceptance for publication. References must be set according to the journal's Instruction for authors. Please check https://www.mdpi.com/journal/medicina/instructions EndNoting them and choosing the MDPI style  is very useful.

References should be described as follows, depending on the type of work:

  • Journal Articles:
    1. Author 1, A.B.; Author 2, C.D. Title of the article. Abbreviated Journal Name Year, Volume, page range.
  • Books and Book Chapters:
    2. Author 1, A.; Author 2, B. Book Title, 3rd ed.; Publisher: Publisher Location, Country, Year; pp. 154–196.
    3. Author 1, A.; Author 2, B. Title of the chapter. In Book Title, 2nd ed.; Editor 1, A., Editor 2, B., Eds.; Publisher: Publisher Location, Country, Year; Volume 3, pp. 154–196.

What is recommended to be Italics - must be italics, what is recommended to be bold - must be bolded, semicolon must be inserted between the names of the authors, etc.

Author Response

To:

Medicina

Cover letter

Dear Editor in Chief,

We thank the editor and the reviewers for their comments and suggestions which enabled us to largely improve our manuscript.

Here enclosed you’ll find the “point to point” answers to comments and questions.

Reviewer 2

The authors made some improvements, but not the all requested. Once again: THE ASPECT of a paper and RESPECTING ALL the Instructions for authors is very important in its submission step. Please see below my concerns (which were also mentioned in my 1st report). I detailed them as follows:

  1. I understand that authors they did not want to respect one of my requests (from my 1st report) regarding:

  • I recommend to insert it in a new section, namely 2. Methodology. A PRISMA flow chart is recommended to be done please see both page at al. papers, where this type of graphic is very well described: Page, M.J.; McKenzie, J.E.; Bossuyt, P.M.; Boutron, I.; Hoffmann, T.C.; Mulrow, C.D.; Shamseer, L.; Tetzlaff, J.M.; Akl, E.A.; Brennan, S.E.; et al. The PRISMA 2020 statement: An updated guideline for reporting systematic reviews. Journal of Clinical Epidemiology 2021, 134, 178-189, doi:10.1016/j.jclinepi.2021.03.001. Page, M.J.; McKenzie, J.E.; Bossuyt, P.M.; Boutron, I.; Hoffmann, T.C.; Mulrow, C.D.; Shamseer, L.; Tetzlaff, J.M.; Moher, D. Updating guidance for reporting systematic reviews: development of the PRISMA 2020 statement. Journal of Clinical Epidemiology 2021, 134, 103-112, doi:10.1016/j.jclinepi.2021.02.003. Please renumber the other sections accordingly.

  • In this methodology part, please mention all criteria used for the papers' selection. You will see all of them in the PRISMA flow chart provided as model in the papers above recommended.

But I still believe that it is the easiest way to present the selection of the references included in the study. I gave the references (which are not mines) exactly to help them in understanding a PRISMA flow chart (which is very easy to be done and followed. I apply it in all my review type papers.

We followed the reviewer's request and added a paragraph and related diagram according to the PRISMA guidelines. This paragraph has been named “2. Methodology”, as suggested by the reviewer.  See lines 116-193.

  1. Table 1 does not respect the Instructions for authors regarding settings. Please check https://www.mdpi.com/journal/medicina/instructions.

According to the referee’s suggestions, in the new version of the article, we followed the Instructions for authors for the tables design.

  1. There remained in the text many very short paragraphs  that must be merged.

In the new version of the article, we merged some shorter paragraphs

  1. Please remove empty spaces between lines in the same section/subsection. Revise the entire manuscript in this regard. (i.e. section 5, 6, etc.)

As suggested by the reviewer, we removed empty spaces between lines (See sections 5 and 6).

  1. In my previous report I also suggested :
  • I suggest to tabulate all data between L264-443. An exhaustive Table is much more relevant and easier to follow, especially when it comprises data from different references. The last column of the table must be named Ref. (from References), and all of them must be mentioned there, as well in brackets. Please check the style of the table recommended by the Instructions for authors. https://www.mdpi.com/journal/medicina/instructions.

We thank the reviewer for this suggestion. We modified the table adding last column named Ref and we also have corrected the style of the table as recommended in the Instructions for authors.

  • Furthermore in this section, please discuss the beneficial roles of PCSK9 inhibitors in DKD, the role of fibrates in DKD, the role of mineralocorticoid receptor inhibitors with non-steroid structure (such as finernone) in the treatment of these patients.

As suggested by the reviewer we discussed these important topics. See line 576-597 and Table 2.

The authors tabulated some of them but left also the text. Data must be provided in a single form (not in duplicate and I requested the table exactly for the reason to remove that the long list of potentially usable therapeutics (again with many empty lines and very short paragraphs for a single therapeutic). So, please complete the table with more details, and remove the text list of therapeutics/drugs. Table 2 does not respect the Instructions for authors as well. Furthermore, it can be "widened", adding more columns (if the authors consider it necessary), so that it occupies the entire width of the page.

The title of a Table must be inserted ABOVE the table.

Last column can be renamed in short as Ref. and is enough to mention there just the number of the reference used to support the text  - i.e. for SGL T2 [152-156]. The authors complicated themselves in this last column.

In the new version of the article, we followed the reviewer's requests.

From my 1st report also. References. remained written at the final of the manuscript very carelessly, in a different style, not respecting any rules. The appearance of a work (in its entirety), as well as its content, is defining in its acceptance for publication. References must be set according to the journal's Instruction for authors. Please check https://www.mdpi.com/journal/medicina/instructions EndNoting them and choosing the MDPI style  is very useful.

References should be described as follows, depending on the type of work:

  • Journal Articles:
    Author 1, A.B.; Author 2, C.D. Title of the article. Abbreviated Journal NameYearVolume, page range.
  • Books and Book Chapters:
    Author 1, A.; Author 2, B. Book Title, 3rd ed.; Publisher: Publisher Location, Country, Year; pp. 154–196.
    3. Author 1, A.; Author 2, B. Title of the chapter. In Book Title, 2nd ed.; Editor 1, A., Editor 2, B., Eds.; Publisher: Publisher Location, Country, Year; Volume 3, pp. 154–196.

What is recommended to be Italics - must be italics, what is recommended to be bold - must be bolded, semicolon must be inserted between the names of the authors, etc.

According to the reviewer’s suggestions in the new correct revision of the manuscript, the references are written according to the journal's instruction for authors.